# Estimation of regional polygenicity from GWAS provides insights into the genetic architecture of complex traits

Ruth Johnson[1]*, Kathryn S. Burch[2], Kangcheng Hou[2], Mario Paciuc[3], Bogdan Pasaniuc[2,4,5,6👁], Sriram Sankararaman[1,2,5,6👁]*

1 Department of Computer Science, University of California, Los Angeles, California, United States of America, 2 Bioinformatics Interdepartmental Program, University of California, Los Angeles, California, United States of America, 3 Department of Statistics, Rice University, Houston, Texas, United States of America, 4 Department of Pathology and Laboratory Medicine, David Geffen School of Medicine, University of California, Los Angeles, California, United States of America, 5 Department of Human Genetics, University of California, Los Angeles, California, United States of America, 6 Department of Computational Medicine, David Geffen School of Medicine, University of California, Los Angeles, California, United States of America

👁 These authors contributed equally to this work.
* ruthjohnson@ucla.edu (RJ); sriram@cs.ucla.edu (SS)

**Data Availability Statement:** The BEAVR software is available as open source software at github.com/bogdanlab/BEAVR.

## Abstract

The number of variants that have a non-zero effect on a trait (*i.e.* polygenicity) is a fundamental parameter in the study of the genetic architecture of a complex trait. Although many previous studies have investigated polygenicity at a genome-wide scale, a detailed understanding of how polygenicity varies across genomic regions is currently lacking. In this work, we propose an accurate and scalable statistical framework to estimate regional polygenicity for a complex trait. We show that our approach yields approximately unbiased estimates of regional polygenicity in simulations across a wide-range of various genetic architectures. We then partition the polygenicity of anthropometric and blood pressure traits across 6-Mb genomic regions ($N$ = 290K, UK Biobank) and observe that all analyzed traits are highly polygenic: over one-third of regions harbor at least one causal variant for each of the traits analyzed. Additionally, we observe wide variation in regional polygenicity: on average across all traits, 48.9% of regions contain at least 5 causal SNPs, 5.44% of regions contain at least 50 causal SNPs. Finally, we find that heritability is proportional to polygenicity at the regional level, which is consistent with the hypothesis that heritability enrichments are largely driven by the variation in the number of causal SNPs.

## Author summary

The proportion of SNPs with nonzero effects on a trait, or polygenicity, is a key quantity used to describe the genetic architecture of a complex trait. Furthermore, identifying the specific genomic regions that contribute to trait variation requires an understanding of how the number of causal SNPs varies across regions of the genome (regional

**Funding:** RJ is partially supported by NSF grant DGE-1829071. SS is supported in part by NIH grant R35GM125055, NSF grant III-1705121, an Alfred P. Sloan Research Fellowship grant FG-2017-9480, and a gift from the Okawa Foundation. BP is supported in part by NIH grants R01HG009120, R01HG006399, and U01CA194393. The funders had no role in study design, data collection and analysis, decision to publish, or preparation of the manuscript.

**Competing interests:** The authors have declared that no competing interests exist.

polygenicity). In this work, we propose a statistical framework to estimate regional polygenicity for a complex trait using marginal effect sizes from GWAS and LD information. We demonstrate in simulation and empirical data that our approach accurately and efficiently estimates regional polygenicity. We find that SNP-heritability is proportional to polygenicity both on the genome-wide and regional scale, suggesting that the observed differences in heritability across traits stem from differences in the underlying number of causal SNPs.

## Introduction

*Polygenicity, i.e.*, the proportion of SNPs with nonzero effects on a trait, is a key quantity in efforts to understand the genetic architecture of complex traits. Accurate estimates of genome-wide polygenicity can be used to improve the prediction accuracy of polygenic risk scores [1, 2], quantify the strength of selection acting on a trait [3, 4], or better understand the biological complexity of the pathways driving disease risk [5, 6]. A major challenge in estimating polygenicity from genome-wide association study (GWAS) data arises due to the correlations between nearby SNPs, *i.e.* linkage disequilibrium (LD). In the presence of LD, methods for estimating polygenicity need to search over all possible causal status configurations at each SNP which, in turn, leads to an intractable computation for regions that harbor even a modest number of SNPs. Several methods implicitly model polygenicity in the context of phenotype prediction [7–11] whereas other methods explicitly aim to estimate polygenicity [2, 3], with recent methods overcoming the computational bottleneck by making simplifying model assumptions about the relationship between LD and polygenicity [2]. While all previous studies have focused on genome-wide polygenicity and its variation across traits [2], identification of genomic regions that are important for trait variation requires an understanding of how the number of causal SNPs varies across the genome (*regional polygenicity*).

In this work, we propose a statistical framework, Bayesian Estimation of Variants in a Region (BEAVR), to estimate regional polygenicity for a complex trait. Our approach estimates the proportion of causal variants in a given region ($p_r$) using marginal effect sizes from GWAS and in-sample LD information. In this work, we define 'causal variants' as a set of variants measured in a given GWAS study that have either a nonzero effect on the trait or tag unmeasured variants through LD that also have a nonzero effect. This particular definition does not imply a causal biological relationship nor formal causation as defined in causal inference. Thus, the estimates of polygenicity are defined with respect to the set of variants in the analyzed GWAS. This is similar to the definition of SNP-heritability estimates which are also specific to each set of variants and cannot be extrapolated to other sets of SNPs [12–15].

The Bayesian model in BEAVR imposes a prior on the true SNP effect sizes where the probability of a non-zero true effect size at each SNP in the region is given by $p_r$ [16, 17]. The observed GWAS effect sizes are obtained as a noisy combination of the unobserved true SNP effect sizes [18, 19]. We use Markov chain Monte Carlo (MCMC) [20] to approximate the posterior probability of the regional polygenicity parameter. This inference problem is computationally challenging as it requires disentangling correlations between SNPs due to LD. Leveraging the insight that the genetic architectures of most traits are likely to be sparse (so that most SNPs are not causal), we obtain a substantially more efficient MCMC algorithm that allows us to infer regional polygenicity across a large number of SNPs.

We validate our approach using extensive simulations and find that our method accurately estimates polygenicity in realistic settings; BEAVR estimates yield a relative bias $< 2\%$ across all simulations whereas existing methods obtain biased estimates particularly in simulations with high degrees of polygenicity (*i.e.* $p_r > 5\%$). Next, we estimate regional polygenicity across 6-Mb regions for five quantitative anthropometric and blood pressure traits in the UK Biobank ($N = 290, 641$ unrelated British individuals) restricting to genotyped SNPs with MAF $> 1\%$. Consistent with previous works [2], we find that all analyzed traits are highly polygenic at the genome-wide scale: over one-third of regions harbor at least one causal SNP across all traits. The proportion of regions containing at least one causal SNP (typically defined as regions with significant heritability) has been used as a proxy for polygenicity in earlier studies [12, 21]; we find that the proportion of regions containing at least one causal SNP is much higher than the estimated polygenicity. For example, while 79.6% of regions contain at least one causal SNP for height, the genome-wide polygenicity is estimated to be 3.07%. Additionally, we observe wide variation in regional polygenicity: on average across all analyzed traits, 48.9% of regions contain at least 5 causal SNPs while 5.44% of regions contain at least 50 causal SNPs, demonstrating the additional information provided from estimates of regional polygenicity. Finally, we find that within traits, regional SNP-heritability is proportional to regional polygenicity, suggesting that variation in heritability across the genome is largely driven by variation in the number of causal SNPs.

## Materials and methods

### Generative model

We assume that the trait measured in individual $i$, $y_i$, is a linear function of standardized genotypes $x_i = (x_{i,1}, \cdots, x_{i,M})$ measured at $M$ SNPs with true SNP effect sizes $\boldsymbol{\beta} = (\beta_1, \cdots, \beta_M)$ and an independent additive noise term $\epsilon_i$.

$$y_i = \sum_{m=1}^{M} \beta_m x_{i,m} + \epsilon_i, i \in \{1, \cdots N\} \tag{1}$$

$$\epsilon_i \overset{iid}{\sim} \mathcal{N}(0, \sigma_e^2) \tag{2}$$

We model a non-infinitesimal trait architecture in which a subset of the $M$ SNPs are causal by imposing a spike-and-slab prior on the causal effect sizes $\boldsymbol{\beta}$ [2, 3, 7]. We represent the causal statuses across the SNPs as $\boldsymbol{c} = (c_1, \cdots, c_M)$. Here, $c_m = 1$ if SNP $m$ is a causal SNP with probability $p$ and 0 otherwise. Thus, $p$ denotes the proportion of causal SNPs or the *polygenicity*.

The Gaussian slab is parametrized with mean 0 and variance $\frac{h_{GW}^2}{Mp}$ where $h_{GW}^2$ is the genome-wide heritability. We draw independent Gaussian random variables for each of the $M$ SNPs: $\boldsymbol{\gamma} = (\gamma_1, \cdots, \gamma_M)$. The effect size $\beta_m$ is $\gamma_m$ if SNP $m$ is causal and 0 otherwise.

$$\gamma_m \sim \mathcal{N}(0, \frac{h_{GW}^2}{Mp}) \tag{3}$$

$$\beta_m \mid c_m, \gamma_m = \begin{cases} \gamma_m & \text{if } c_m = 1 \\ 0 & \text{if } c_m = 0 \end{cases} \tag{4}$$

We model the conditional distribution of the GWAS effect sizes given the true effect sizes where $\hat{\beta}_m$ is the estimated marginal effect size of SNP $m$ for the trait.

$$\hat{\boldsymbol{\beta}} | \boldsymbol{\beta} \sim \mathcal{N}(\boldsymbol{V}\boldsymbol{\beta}, \boldsymbol{V}\sigma_e^2) \tag{5}$$

Here the covariance matrix is parametrized by the environmental noise $\sigma_e^2$ and the correlations among SNPs, *i.e.* the linkage disequilibrium (LD) matrix $\boldsymbol{V}$. The variance of the environmental noise term is parameterized by $\sigma_e^2 = \frac{1 - h_{GW}^2}{N}$, where $N$ is the number of individuals in the study.

We impose a symmetric Beta prior on the polygenicity parameter, $p$.

$$p \sim Beta(\alpha, \alpha) \tag{6}$$

In practice, we use $\alpha = 0.2$ to put a higher weight on the tails of the Beta distribution. The full graphical model can be found in S1 Fig.

In this work, we focus on accurately estimating the proportion of causal variants in a given region $r$ (regional polygenicity, $p_r$). We assume that the above proposed genome-wide generative model holds when applied only within a specific region of the genome. This includes modeling the heritability only within that region ($h_r^2$) instead of the genome-wide heritability ($h_{GW}^2$). Modeling each region separately also assumes that there are no correlations between regions, such as correlations due to long-range LD. This assumption is reasonable when regions are chosen to correspond to LD blocks or when regions are sufficiently large such that correlations with adjacent regions may be ignored. Therefore, the LD matrix used in the regional model would only be the LD computed from SNPs within that particular region ($\boldsymbol{V}_r$). Additionally, although our framework naturally estimates the SNP effect sizes and posterior inclusion probabilities (*i.e.*, the probability that a given SNP is causal), we focus in this work on the posterior probability of $p_r$.

The posterior probability of the model parameters of interest ($p_r, \boldsymbol{\gamma}_r, \boldsymbol{c}_r$) for a given region $r$ is given by:

$$P(p_r, \boldsymbol{\gamma}_r, \boldsymbol{c}_r \mid \hat{\boldsymbol{\beta}}_r, \alpha, h_r^2) \propto P(p_r \mid \alpha)P(\boldsymbol{c}_r \mid p_r)P(\boldsymbol{\gamma}_r \mid h_r^2, p_r)P(\hat{\boldsymbol{\beta}}_r \mid \boldsymbol{\gamma}_r, \boldsymbol{c}_r, h_r^2) \tag{7}$$

## Inference

We use Markov Chain Monte Carlo (MCMC) to approximate the posterior probability as defined in Eq 7. Specifically, we derive a Gibbs sampler [22] to sample from the posterior distribution of the regional polygenicity $p_r$ and latent variables ($\boldsymbol{c}_r, \boldsymbol{\gamma}_r$). The method takes as input the marginal effect sizes from GWAS for a single trait in a region $r$ ($\hat{\boldsymbol{\beta}}_r$), the matrix of SNP correlations or LD per region ($\boldsymbol{V}_r$), an estimate of the SNP heritability in that region ($h_r^2$), and the sample size of the GWAS ($N$). As output, we estimate the posterior probability of the regional polygenicity for region $r$ ($p_r$).

**Transforming GWAS effect sizes.** To facilitate efficient inference, we transform the marginal effects from GWAS: $\tilde{\boldsymbol{\beta}}_r \equiv \boldsymbol{V}_r^{-\frac{1}{2}} \boldsymbol{\beta}_r$. The conditional probability of these transformed effects is given by:

$$\tilde{\boldsymbol{\beta}}_r \mid \boldsymbol{\beta}_r \sim \mathcal{N}\left(\boldsymbol{V}_r^{\frac{1}{2}} \boldsymbol{\beta}_r, \boldsymbol{I}_{M_r}\sigma_e^2\right)$$

Here, $\boldsymbol{I}_{M_r}$ is the identity matrix of size $M_r \times M_r$ where $M_r$ is the number of SNPs in region $r$. We note that this is a one-time transformation that is performed before running the sampler. These transformed effects can be efficiently computed and stored for each genomic region.

**Sampling $\gamma_r$ and $c_r$.** We recall that the true effect size at SNP $m$ in region $r$ ($\beta_{r,m}$) is given by a spike-and-slab prior parametrized by the causal effect size ($\gamma_{r,m}$) and the causal status at that SNP ($c_{r,m}$) (see Eq 4). We choose to sample $\gamma_{r,m}$ and $c_{r,m}$ together in a block step in the Gibbs sampler update.

Let $\boldsymbol{\theta}_r = \{(\boldsymbol{\gamma}_{\neg r,m}, \boldsymbol{c}_{\neg r,m}), h_r^2, p_r, \alpha\}$, where $\boldsymbol{\gamma}_{\neg r,m}$ denotes all effect sizes except for the effect of the $m^{th}$ SNP; this similarly follows for $\boldsymbol{c}_{\neg r,m}$.

$$P(\gamma_{r,m}, c_{r,m} \mid \boldsymbol{\theta}_r, \tilde{\boldsymbol{\beta}}_r) = P(\gamma_{r,m} \mid c_{r,m}, \boldsymbol{\theta}_r, \tilde{\boldsymbol{\beta}}_r)P(c_{r,m} \mid \boldsymbol{\theta}_r, \tilde{\boldsymbol{\beta}}_r)$$

We are interested in the posterior probability of the causal effect size $\gamma_{r,m}$ when $c_{r,m} = 1$ since $P(\gamma_{r,m} \mid c_{r,m} = 0) = 0$ due to the spike-and-slab prior. This can be expressed as:

$$P(\gamma_{r,m} \mid c_{r,m} = 1, \boldsymbol{\theta}_r, \tilde{\boldsymbol{\beta}}_r) \propto P(\tilde{\boldsymbol{\beta}}_r \mid \gamma_{r,m}, c_{r,m} = 1, \boldsymbol{\theta}_r)P(\gamma_{r,m} \mid c_{r,m} = 1, \boldsymbol{\theta}_r)$$

Working with the transformed GWAS effect sizes, the posterior distribution of $\gamma_{r,m}$ becomes univariate Gaussian with the following mean and variance. Here we denote $\boldsymbol{r}_{r,m} = \tilde{\boldsymbol{\beta}}_r - \boldsymbol{V}_r^{\frac{1}{2}}\boldsymbol{\gamma}_r \circ \boldsymbol{c}_r + \boldsymbol{V}_{r,m}^{\frac{1}{2}}\gamma_{r,m}c_{r,m}$, which is the residual from subtracting the effects of all SNPs except for SNP $m$ (here $\boldsymbol{V}_{r,m}^{\frac{1}{2}}$ denotes column $m$ of the matrix $\boldsymbol{V}_r^{\frac{1}{2}}$). We define $\sigma_{r,g}^2 = \frac{h_r^2}{M_r p_r}$ and $\sigma_e^2 = \frac{1-h_r^2}{N}$ for the region-specific model. See S1 File for full derivation details.

$$P(\gamma_{r,m} \mid c_{r,m} = 1, \boldsymbol{\theta}_r, \tilde{\boldsymbol{\beta}}_r) = \mathcal{N}(\gamma_{r,m}; \mu_{r,m}, \sigma_{r,m}^2)$$

$$\frac{1}{\sigma_{r,m}^2} = \frac{1}{\sigma_{r,g}^2} + \frac{1}{\sigma_e^2}\boldsymbol{V}_{r,m}^{\frac{1}{2}\top}\boldsymbol{V}_{r,m}^{\frac{1}{2}} \qquad (8)$$

$$\mu_{r,m} = \sigma_{r,m}^2 \frac{1}{\sigma_e^2}\boldsymbol{r}_{r,m}^{\top}\boldsymbol{V}_{r,m}^{\frac{1}{2}}$$

We sample $c_{r,m} \mid \boldsymbol{\theta}_r, \tilde{\boldsymbol{\beta}}_r$ from a Bernoulli distribution with parameter $P(c_{r,m} = 1 \mid \boldsymbol{\theta}_r, \tilde{\boldsymbol{\beta}}_r)$:

$$
\begin{aligned}
P(c_{r,m} = 1 \mid \boldsymbol{\theta}_r, \tilde{\boldsymbol{\beta}}_r) &= \int P(c_{r,m} = 1, \gamma_{r,m} \mid \boldsymbol{\theta}_r, \tilde{\boldsymbol{\beta}}_r)d\gamma_{r,m} \\
&= \int \frac{P(\tilde{\boldsymbol{\beta}}_r \mid \gamma_{r,m}, c_{r,m} = 1, \boldsymbol{\theta}_r)P(\gamma_{r,m}, c_{r,m} = 1 \mid \boldsymbol{\theta}_r)}{P(\tilde{\boldsymbol{\beta}}_r \mid \boldsymbol{\theta}_r)}d\gamma_{r,m} \\
&= \frac{(p_r)\sqrt{\frac{\sigma_{r,m}^2}{\sigma_{r,g}^2}}\exp\{\frac{1}{2\sigma_{r,m}^2}\mu_{r,m}^2\}}{(p_r)\sqrt{\frac{\sigma_{r,m}^2}{\sigma_{r,g}^2}}\exp\{\frac{1}{2\sigma_{r,m}^2}\mu_{r,m}^2\} + (1 - p_r)} \\
&= d_{r,m}
\end{aligned}
$$

**Sampling $p_r$.** The complete conditional posterior distribution of $p_r$ depends not only on the causal status of each SNP ($c_{r,m}$), but also on the latent variable ($\gamma_{r,m}$) since $p_r$ parametrizes the variance term of $\gamma_{r,m}$. We sample from this distribution using a random-walk Metropolis-Hastings step [20]. We use a Beta distribution as a proposal distribution:

$$
\begin{aligned}
p_r^* &\sim Q(p_r^* \mid p_r) \\
&= Beta(\alpha + Cp_r, \alpha + C(1 - p_r))
\end{aligned}
$$

Here, $C$ is a constant that controls the variance of the proposal distribution. In practice we found that $C = 10$ yield effective mixing.

## Leveraging sparsity of the genetic architecture to improve the computational efficiency

The key computational bottleneck in the Gibbs sampling scheme involves computing the mean of the posterior distribution of the causal effect size at SNP $m$ ($\mu_{r,m}$ in Eq 8). Specifically, the matrix computations associated with the residual term, $\boldsymbol{r}_{r,m} = \tilde{\boldsymbol{\beta}}_r - \boldsymbol{V}_r^{\frac{1}{2}}\boldsymbol{\gamma}_r \circ \boldsymbol{c}_r + \boldsymbol{V}_{r,m}^{\frac{1}{2}}\gamma_{r,m}c_{r,m}$, naively scales as $\mathcal{O}(M_r^2)$ due to the middle term, which is a matrix of size $M_r \times M_r$ multiplied by a vector of size $M \times 1$. Because this computation must be performed for every SNP, the overall complexity of the sampler is $\mathcal{O}(M_r^3)$ if implemented in this straightforward fashion. Below, we will break down the posterior mean term such that the complexity of computing $\boldsymbol{r}_{r,m}$ will only be $\mathcal{O}(K_r)$, where $K_r$ is the number of causal SNPs in the region, and the complexity of the sampler will be $\mathcal{O}(K_r M_r)$. This is accomplished by two steps: i) breaking the equation into constant terms that do not need to be updated at every iteration of the sampler, ii) leveraging the expected sparsity of the true causal vector and only performing computations over the causal SNPs.

Writing out the posterior mean term and expanding, we have:

$$
\begin{aligned}
\mu_{r,m} &= \frac{\sigma_{r,m}^2}{\sigma_e^2}\,\boldsymbol{r}_{r,m}^\top\,\boldsymbol{V}_{r,m}^{\frac{1}{2}} \\
&= \frac{\sigma_{r,m}^2}{\sigma_e^2}\left[\tilde{\boldsymbol{\beta}}_r - \boldsymbol{V}_r^{\frac{1}{2}}\boldsymbol{\gamma}_r \circ \boldsymbol{c}_r + \boldsymbol{V}_{r,m}^{\frac{1}{2}}\gamma_{r,m}c_{r,m}\right]^\top\boldsymbol{V}_{r,m}^{\frac{1}{2}} \\
&= \frac{\sigma_{r,m}^2}{\sigma_e^2}\left[\tilde{\boldsymbol{\beta}}_r - \sum_{m\neq l}^{M_r}\boldsymbol{V}_{r,l}^{\frac{1}{2}}\boldsymbol{\gamma}_{r,l}\boldsymbol{c}_{r,l}\right]^\top\boldsymbol{V}_{r,m}^{\frac{1}{2}} \\
&= \frac{\sigma_{r,m}^2}{\sigma_e^2}\left[\tilde{\boldsymbol{\beta}}_r^\top\boldsymbol{V}_{r,m}^{\frac{1}{2}} - \sum_{l\neq m,c_{r,l}=1}^{M_r}\boldsymbol{V}_{r,l}^{\frac{1}{2}\top}\boldsymbol{V}_{r,m}^{\frac{1}{2}}\gamma_{r,l}c_{r,l}\right]
\end{aligned}
$$

The first term, $\tilde{\boldsymbol{\beta}}_r^\top\boldsymbol{V}_{r,m}^{\frac{1}{2}}$, is composed of the vector of GWAS effect sizes and a vector of the LD matrix corresponding to the $m^{th}$ SNP, neither of which are updated within the sampler. Second, the term $\boldsymbol{V}_{r,l}^{\frac{1}{2}}\boldsymbol{V}_{r,m}^{\frac{1}{2}}$ can also be pre-computed since it is only the product of two columns within the LD matrix. Aside from the variance terms at the beginning of the equation, which are only scalars, the only term that varies at each iteration of the sampler is $\gamma_{r,l}\,c_{r,l}$ since both the effect size and causal status need to be re-sampled at each iteration. Since this term is wrapped in a summation over $M_r$ SNPs, the complexity of computing $\mu_{r,m}$ is currently $\mathcal{O}(M_r)$. However, even with this simplification, the overall complexity of the sampler is $\mathcal{O}(M_r^2)$ since this mean term must be computed at every SNP at every iteration.

To further simplify the computation, we can leverage the observation that most complex traits contain only a small proportion of causal SNPs ($K_r$) in each region. As the sampler converges to the stationary distribution, we would expect the causal status vector ($\boldsymbol{c}_r$) to be sparse, where $K_r \ll M_r$. When this occurs, the summation term will only include a few non-zero terms. By only subtracting the non-zero terms, this term is simply reduced to the number of causal variants and the complexity becomes $\mathcal{O}(K_r)$. Even though this computation must be

done at each SNP, the overall complexity of the sampler is only $\mathcal{O}(K_r M_r)$ which is tractable under the assumption of $K_r \ll M_r$.

## Simulation analysis

**Simulations for marginal effects using LD information.** Using pre-computed LD information, we generated marginal effect sizes for a given region from synthetic GWAS that reflect a variety of genetic architectures. We denote the number of SNPs in a region as $M_r$ and the regional polygenicity as $p_r$. We denote the causal indicator status of each SNP in each region as $c_{r,m} \in \{0, 1\}$, where $c_{r,m} = 1$ if the $m^{th}$ SNP is causal and 0 otherwise for $m = 1, \cdots, M_r$ and regions $r = 1, \cdots, R$.

The causal status of a SNP is generated from:

$$c_{r,m} \sim Ber(p_r)$$

If $c_{r,m} = 1$, the effect size of SNP $m$ within the $r^{th}$ region is drawn from a univariate Gaussian distribution with mean 0 and variance equal to the regional heritability $(h_r^2)$ divided by the number of casual SNPs:

$$\beta_{r,m} \sim \begin{cases} 0, & c_{r,m} = 0, \\ \mathcal{N}\left(0, \frac{h_r^2}{M_r p_r}\right), & c_{r,m} = 1 \end{cases}$$

Marginal association statistics for the region are then generated from the following model:

$$\hat{\boldsymbol{\beta}}_r \mid \boldsymbol{\beta}_r \sim \mathcal{N}(\boldsymbol{V}_r \boldsymbol{\beta}_r, \boldsymbol{V}_r \sigma_e^2)$$

Here, the environmental noise is a function of the sample size and heritability of the trait, $\sigma_e^2 = \frac{1 - h_r^2}{N}$. We use regional LD computed with genotypes from 337, 205 unrelated (less related than third-degree relatives), white, British individuals ($M_r = 1, 000$ array SNPs) from the UK Biobank [23]. The LD matrix for a region is computed as $\boldsymbol{V}_r = \frac{X_r^\top X_r}{N}$, where $\boldsymbol{X}_r$ is the genotype matrix using only SNPs within region $r$.

Using the framework above, we generated marginal effect sizes where we varied the regional polygenicity from $p_r = 0.005, 0.01, 0.05,$ and $0.10$, genome-wide heritability from $h_{GW}^2 = 0.10, 0.25,$ and $0.50$, and the sample size from $N = 50K, 500K, 1M$ individuals, which is comparable to the sample sizes of many current GWAS studies [24, 25]. For each simulated region, we set the number of SNPs per region to 1, 000. For the regional heritability parameter, we used the simulated genome-wide heritability scaled by the number of SNPs in the region, $M_r$, and the number of SNPs on the array, $M$: $h_r^2 = \frac{h_{GW}^2 M_r}{M}$.

To estimate the regional polygenicity, we ran BEAVR for 1,000 iterations with a burn-in of 250 iterations. We used the same LD information that was used for simulation (*i.e.* "in-sample" LD). We also computed regional polygenicity using GENESIS [2]. We ran GENESIS using the default parameter settings and LD information from 1000 Genomes [26]. We used both the 2-component and 3-component settings when running GENESIS. We note that the implementation of GENESIS uses the 1000 Genomes LD matrix as a default and there is no option to specify an alternative LD matrix. We averaged the performance of each method across 100 replicates.

**Simulations for marginal effects computed from individual genotype and phenotype data.** Using SNP data ($M = 9, 564$ array SNPs from chromosome 22, $N = 337K$ individuals) from a group of unrelated, self-identified British, white ancestry individuals from the UK

Biobank [23], we simulated marginal effects by generating phenotypes from real genotype array data. For this analysis, the set of unrelated individuals is defined as pairs of individuals with kinship coefficient $< \frac{1}{2}^{(9/2)}$ (greater than third-degree relatives) [23]. Then we performed ordinary least squares to estimate the marginal effect size of each SNP. Given the standardized genotype matrix $X$ and the genome-wide SNP heritability $h^2_{GW}$, phenotypes are generated as follows.

We set the genome-wide proportion of causal variants to be $p = 0.01$. We denote the causal indicator status of each SNP as $c_m \in \{0, 1\}$, where $c_m = 1$ if the $m^{th}$ SNP is causal and 0 otherwise for $m = 1, \cdots, M$. Standardized effects and phenotypes are generated from the following model. Note that $\sigma^2_m = 0$ if $c_m = 0$.

$$\sigma^2_m = c_m \frac{h^2_{GW}}{Mp}$$

$$(\beta_1, \cdots, \beta_M)^\top \sim \mathcal{N}(0, \text{diag}(\sigma^2_1, \cdots, \sigma^2_M))$$

$$(y_1, \cdots, y_N)^\top \mid \boldsymbol{\beta} \sim \mathcal{N}(X\boldsymbol{\beta}, (1 - h^2_{GW})I_N)$$

Finally, given the phenotypes for all individuals, $\boldsymbol{y} = (y_1, \cdots, y_N)^\top$ and genotypes $X = (\boldsymbol{x}_1^\top, \cdots, \boldsymbol{x}_N^\top)^\top$, we compute marginal association statistics through the OLS estimator, $\hat{\boldsymbol{\beta}} = \frac{1}{N} X^\top \boldsymbol{y}$.

We generated 100 sets of marginal effect sizes where we fixed $p = 0.01$ and $h^2_{GW} = 0.50$. We then estimated the regional polygenicity within each 6-Mb window for chromosome 22 (M = 9,564 array SNPs) using BEAVR. This windowing formed 6 consecutive regions. We used HESS (Heritability Estimator from Summary Statistics) [12], a method for estimating regional heritability at a single region from GWAS summary statistics, to estimate the regional heritability which is then used as input for BEAVR. HESS is run with all default parameters and the same LD matrices used in the simulation framework (*i.e.* in-sample LD). We finally ran BEAVR for 1,000 iterations with a burn-in of 250 iterations and using the same LD information that was used for simulation.

## Analysis of UK Biobank phenotypes

We estimated the partitioned polygenicity for five complex traits in the UK Biobank [23] across 6-Mb windows. We limited our analyses to unrelated individuals with self-identified British, white ancestry. Here, the set of unrelated individuals is defined as pairs of individuals with kinship coefficient $< \frac{1}{2}^{(9/2)}$ (greater than third-degree relatives) [23]. We additionally excluded individuals with putative sex chromosome aneuploidy. All genotypes were standardized, where for each SNP $m$ and individual $n$, we computed $x_{nm} = (g_{nm} - 2f_m)/\sqrt{2f_m(1 - f_m)}$, where $g_{nm} \in \{0, 1, 2\}$ is the number of minor alleles and $f_m$ is the in-sample minor allele frequency (MAF). We then used PLINK [27] (https://www.cog-genomics.org/plink2) to exclude SNPs with MAF $< 0.01$, genotype missingness $> 0.01$, and SNPs that fail the Hardy-Weinberg test at significance threshold $10^{-7}$. We obtained a final set of $N = 290, 641$ individuals for our analyses.

Marginal association statistics were computed through OLS using PLINK. Age, sex, and the top 20 genetic PCs were used as covariates in the regression, where these top 20 PCs were precomputed by the UK Biobank from a superset of 488, 295 individuals. Additional covariates were used for waist-to-hip ratio (adjusted for body mass index (BMI)) and diastolic/systolic blood pressure (adjusted for cholesterol-lowering medication, blood pressure medication, insulin, hormone replacement therapy, and oral contraceptives).

The genome is then divided into 6-Mb windows. Using HESS [12], we estimated the regional heritability within each window for each trait. HESS is run with all default parameters specified and in-sample LD. Using BEAVR and the computed regional heritability estimates, we estimated the regional polygenicity in each 6-Mb window. To initialize the MCMC sampler, we must set initial values for the vector of causal statuses, causal effect sizes, and regional polygenicity ($c_r, \gamma_r, p_r$). For each SNP $m$, if the z-score estimated from GWAS is $\geq 3.5$, then $c_{r,m}$ is initialized to 1 and 0 otherwise. Each causal effect size is drawn from the prior distribution (see Eq 3). The initial value of $p_r$ is set to the proportion of 1's in the initialized causal status vector. We ran the Gibbs sampler for 1, 000 iterations and the first 250 samples were discarded as burn-in. For each region, we computed the posterior mean and posterior standard deviation for $p_r$ from the MCMC samples.

### Annotations in regression analysis

We performed a multivariate regression of the heritability on the estimated number of SNPs from BEAVR, the number of causal SNPs, and genomic annotations within a region. The genomic annotations include the number of genes, median $B$ value (a measure of background selection), and functional annotations [28]. We computed the number of protein coding genes within a region using the protein coding gene sets that have been defined in previous work [29]. If a gene body overlapped two regions, we included the presence of the gene in both regions. Using previously computed $B$ values [30], we computed the median $B$ value of all the SNPs in a region. This quantity was used as the annotation value for that particular region. We additionally included a combination of binary and continuous functional annotations [28]. For each region, we computed the median annotation value for continuous annotations and the proportion of variants with a binary annotation.

## Results

### Simulations

We compare BEAVR to GENESIS [2], an approach that employs a spike-and-slab mixture model to capture both large and small effect sizes at causal SNPs in order to estimate polygenicity at a genome-wide scale (see Materials and methods). To be applicable in genome-wide settings, GENESIS assumes that LD patterns are independent of the probability of a SNP belonging to different mixture components which, in turn, leads to a scalable algorithm. As shown in Fig 1, BEAVR obtains approximately unbiased estimates of polygenicity across each scenario (relative bias < 2% across the simulations). Both the two and three mixture component models from GENESIS obtain relatively unbiased estimates when the true polygenicity is low but demonstrates a severe downward bias in the high polygenicity setting (relative bias > 70% when $p_r = 0.10$). This observation is consistent with our hypothesis that not fully modeling LD limits the ability of GENESIS to accurately estimate parameters, consistent with previously reported downward bias when GENESIS was run with external LD information [2].

Next, we assessed the robustness of our approach to sample size and heritability. We vary the genome-wide heritability to be 0.10 and 0.25 and the sample size to be 50K and 1 million individuals (Fig 2) to fully explore the limitations of our method. We note that when the regional polygenicity $p_r$ is high, BEAVR demonstrates a downward bias either when sample sizes are relatively small ($N = 50K$ individuals) (relative bias 56% and 80% for $p_r = 0.05$ and $p_r = 0.10$ when $h^2_{GW} = 0.50$) or when the heritability is low ($h^2_{GW} = 0.10$) (relative bias 54% and 73% for $p_r = 0.05$ and $p_r = 0.10$ for $N = 500K$). These biases likely arise due to the causal effect sizes being similar in magnitude to the environmental noise, making it difficult to

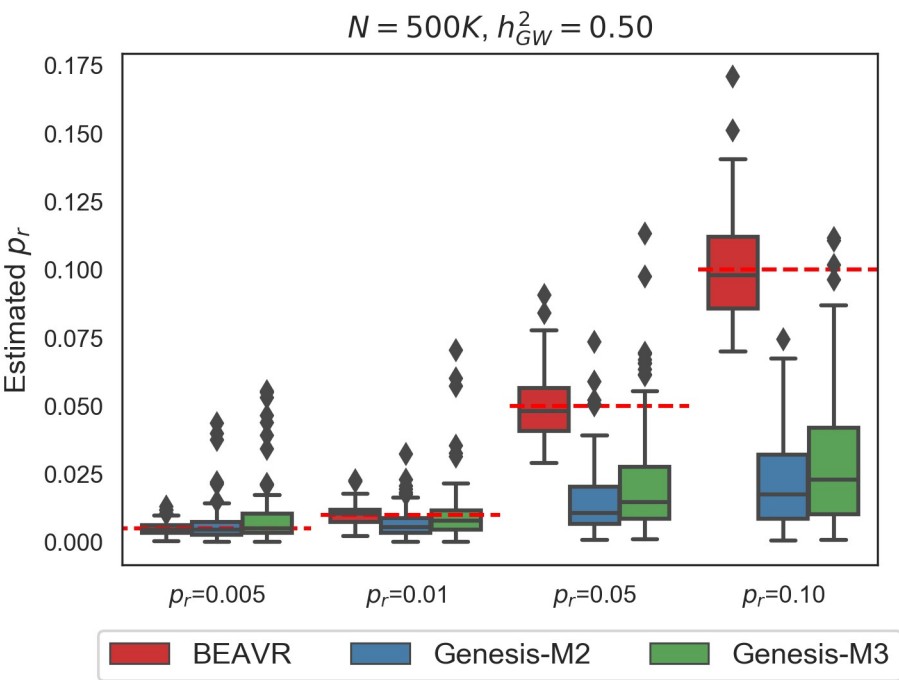

**Fig 1. BEAVR is relatively unbiased in simulated data.** We ran 100 replicates ($M = 1,000$ SNPs, $N = 500$K individuals) where the genome-wide heritability was set to $h^2_{GW} = 0.5$ and the true polygenicity of the region was $p_r = 0.005, 0.01, 0.05, 0.10$. We compared BEAVR to GENESIS-M2 and GENESIS-M3 which employs a spike-and-slab model with either 2 or 3 components (point-mass and either 1 or 2 slabs). All methods are unbiased when the polygenicity is low ($p_r = 0.005, 0.01$). However, when polygenicity is higher ($p_r = 0.05, 0.10$), both GENESIS-M2 and GENESIS-M3 are severely downward biased whereas BEAVR provides unbiased estimates across all settings. Dashed red lines denote true regional polygenicity values in each setting.

correctly identify the causal status of a SNP. Thus, we recommend applying BEAVR to heritable traits measured in large sample sizes.

Next, we performed simulations where GWAS marginal effects are computed from phenotypes simulated from individual genotypes and the regional heritability is estimated directly from the data. Specifically, we simulate phenotypes using individual genotypes for $N = 337$K individuals from the UK Biobank. Each phenotype is simulated to have $h^2_{GW} = 0.50$ and polygenicity $p = 0.01$. We limit our simulations to SNPs from chromosome 22 ($M = 9,564$ SNPs) as each chromosome would be analyzed separately in real data analyses. We then estimate the marginal effect sizes. We divide the simulated data into consecutive regions of 6-Mb for a total of 6 regions, where each region contains 1,000 SNPs on average. We use estimates of regional heritability from GWAS marginal effects (using HESS [12]; see Materials and methods) as input to BEAVR. We find that BEAVR obtains relatively unbiased estimates of polygenicity across all regions (Fig 3A; relative bias < 2% across simulations). These simulations indicate that the polygenicity estimates obtained by BEAVR are robust to heritability estimates that are used as input as well as when LD spans regions. The LD does not significantly affect the estimates likely because the correlation due to LD tends to diminish with genomic distance.

We also explored the robustness of BEAVR to the number of SNPs in the region. Using a simulated GWAS with genome-wide heritability $h^2_{GW} = 0.50$, sample size $N = 500$K, and polygenicity $p_r = 0.01$, we vary the size of the region from $M_r = 500, 1$K, 5K SNPs. From Fig 3B, we can see that the estimates of $p_r$ tend to be unbiased across regions of various sizes although the standard errors tend to increase in smaller regions (relative bias 13%, 1%, and

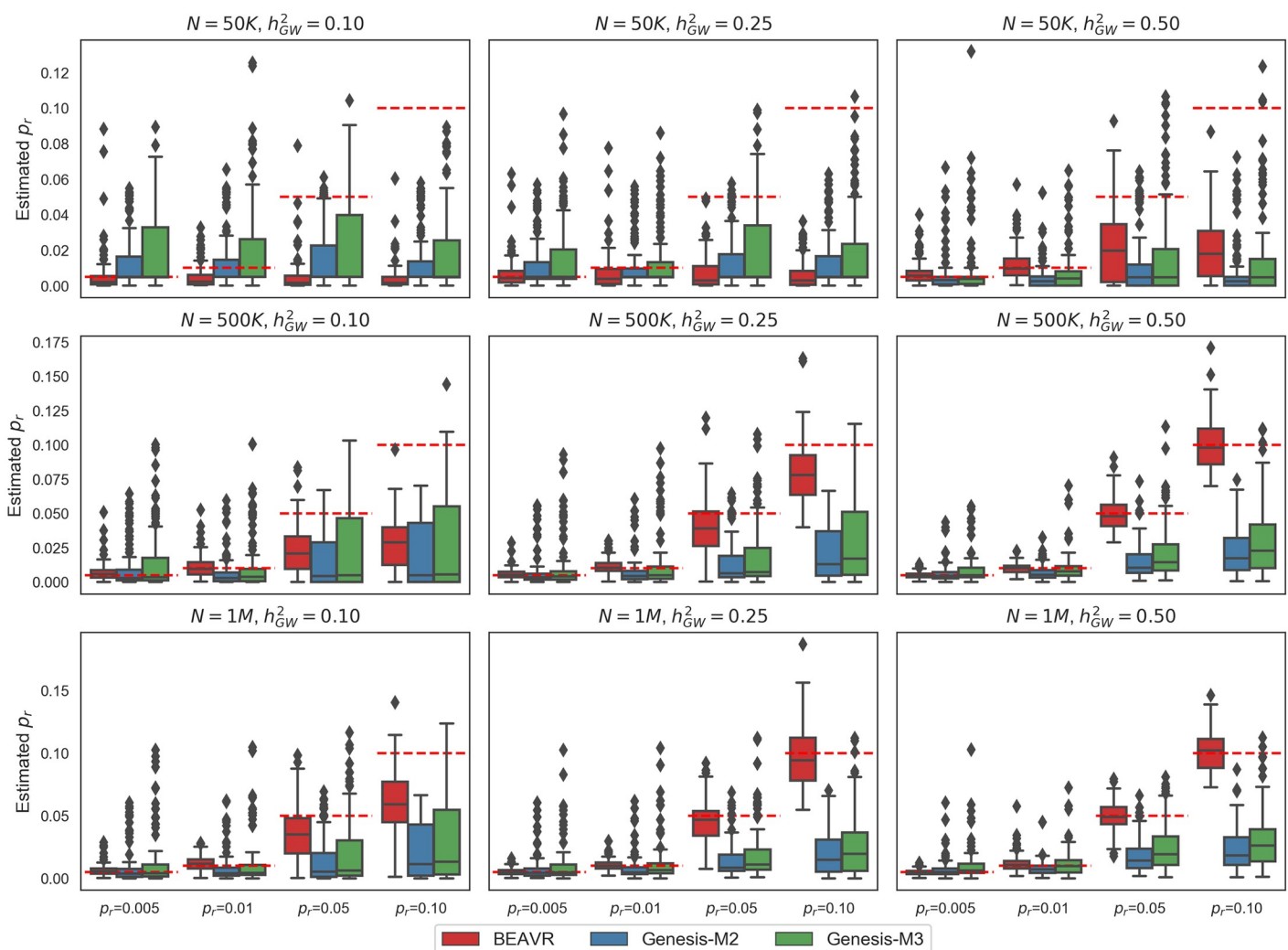

**Fig 2. BEAVR is relatively unbiased across various genetic architectures.** We ran 100 replicates where we vary the genome-wide heritability to be $h^2_{GW} = 0.10$, 0.25, 0.5, the polygenicity of the region to be $p_r = 0.005$, 0.01, 0.05, 0.10, and the sample size $N = 50K$, 500K, 1 million individuals. We compared BEAVR to GENESIS-M2 (2-component) and GENESIS-M3 (3-component). The x-axis denotes the simulated values for the regional polygenicity and the y-axis denotes the estimated values across 100 replicates. Dashed red lines denote the true regional polygenicity value in each setting.

1.2% for $M_r = 500$, 100, 5000 SNPs). This trend occurs because larger regions have a higher number of SNPs to inform the posterior distribution, meaning that there will be higher certainty in the posterior estimates. Additionally, if a region is small, there is a larger impact on the estimated polygenicity when misidentifying causal SNPs due to the small denominator of SNPs in the region. For example, misidentifying a single causal SNP from a set of 10 SNPs will have a greater impact on the bias of polygenicity estimates compared to a set of 1,000 SNPs. These results suggest that BEAVR could potentially be applied to regions of varying length and be used to estimate regional polygenicity around genes or within larger LD blocks.

We next assess the sensitivity of our results when using different hyper-parameters for our prior on the polygenicity parameter $p_r$. Using a simulated GWAS with genome-wide heritability $h^2_{GW} = 0.50$, sample size $N = 500K$, and polygenicity $p_r = 0.01$, we vary our choice of hyper-parameter for the prior on $p_r$: $\alpha = 0.2$, 1, 2. We find that the accuracy of our results is relatively robust to the choice of prior (Fig 3C); we use $\alpha = 0.2$ for all subsequent analyses.

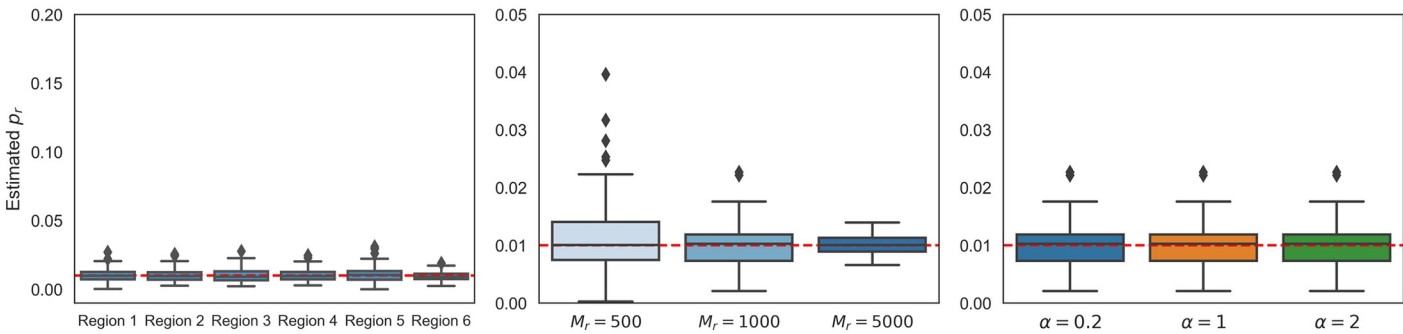

**Fig 3. BEAVR is robust in realistic settings. (A)** Using SNP data from chromosome 22 ($M = 9$, 564 array SNPs, $N = 337K$ individuals), we simulated 100 replicates where the genome-wide heritability was $h^2_{GW} = 0.50$ and $p = 0.01$. We divided the data into 6-Mb consecutive regions for a total of 6 regions and estimated the regional heritability using external software (HESS [12]). Using BEAVR and the estimated regional heritability, we estimated the regional polygenicity to be unbiased across all regions. **(B)** We ran 100 replicates where the genome-wide heritability is fixed $h^2_{GW} = 0.50$, polygenicity $p_r = 0.01$, sample size $N = 500K$, and then varied the number of SNPs in the region from $M = 500$, 1K, 5K SNPs. We used BEAVR to estimate the polygenicity in each region and found our results to be unbiased across all regions. **(C)** We set the genome-wide heritability to $h^2_{GW} = 0.50$, regional polygenicity $p_r = 0.01$, and sample size $N = 500K$. We find that the accuracy of our results is invariant to our choice of prior hyper-parameter ($\alpha$).

Although we assume that causal effect size distributions follow a Gaussian distribution, there are likely traits which do not follow this assumption. We next evaluate the performance of BEAVR when the true causal effect sizes deviate from the assumption of normality. In S2 Fig, we show the distribution of our estimates of regional polygenicity across 3 causal effect size distributions defined by mixture of Gaussian distributions with the following set of variance components and mean 0: $[1 \times 10^{-3}, 1 \times 10^{-4}]$; $[1 \times 10^{-4}, 1 \times 10^{-5}]$; $[1 \times 10^{-3}, 1 \times 10^{-4}, 1 \times 10^{-5}]$. We assume a sample size of $N = 500K$, total regional polygenicity $p_r = 0.01$, and assume the true heritability of the region is known. The number of causal SNPs are spread equally amongst all the mixture components. We see that for causal effect sizes drawn from the distribution with larger variances components (*e.g.* $1 \times 10^{-3}$), our estimates are relatively unbiased. However, for distributions with smaller variance components (*e.g.* $1 \times 10^{-5}$), we start to see a downward bias proportional to the fraction of SNPs drawn from the distribution with the smaller variance component(s). Thus, it is not necessarily the exact shape of the distribution of effect sizes, but the magnitude of the causal effect sizes, which affects the accuracy of the estimates.

## Effect of out-of-sample LD

Although we recommend using in-sample LD when computing estimates of regional polygenicity, we also investigate the scenario where only LD derived from a reference panel is available. We simulate two scenarios: i) reference panel LD is computed from genotypes from individuals of a similar continental population as the target GWAS population but from a separate study (*e.g.* European ancestry individuals from the 1000 Genomes Project); ii) reference panel LD is computed with genotypes from a specific cohort/study and the target GWAS is also conducted with a subset of data from the same the cohort/study or a different version of the study (e.g. 'White, British' individuals from the UK Biobank). This second scenario closely reflects situations where many groups separately apply for freezes of data from the same study yet share GWAS summary statistics across applications.

We simulate the first scenario by simulating 1,000 GWAS regions with $M = 1,000$ SNPs, regional polygenicity $p_r = 0.01$, regional heritability, $h^2_r = 0.0001$ (corresponding to a genome-wide heritability of 0.50), a sample size of $N = 500K$, and use a LD matrix computed from 337, 205 genotypes from unrelated individuals within the 'White, British' population from the UK

Biobank [23]. However, inference is then performed using a reference panel derived LD matrix computed from 503 European ancestry individuals from the 1000 Genomes Project [26]. We find that when using LD from these separate studies, BEAVR fails to accurately estimate the regional polygenicity (S3 Fig). Although the reference panel is constructed using individuals of European ancestry, these individuals were sampled from multiple subcontinental ancestries in Europe (*e.g.* Italy, Spain, Finland). In comparison, the target GWAS population from the UK Biobank is ancestrally homogeneous since it is limited to 'White, British' individuals within the UK.

The second scenario uses the same simulation parameters as above, except the GWAS effect sizes are computed using a LD matrix derived from 168,602 individuals from the unrelated, 'White, British' population within the UK Biobank. Inference is then performed using LD estimated from a separate, non-overlapping set of 168,602 individuals also from the unrelated, 'White, British' population within the UK Biobank. When using LD computed from a separate set of individuals from the same study, we find that our estimates are approximately unbiased (S3 Fig). Our findings show that one can perform inference using a reference panel constructed from a separate set of individuals than used in the GWAS when both sets of individuals are from the same study (*e.g.* UK Biobank). These findings suggest that LD reference panels cannot solely be matched based on the continental ancestry level but need to be matched on a much finer scale. Additionally, differences in study designs between the genotypes used for the LD reference panel and the genotypes used when performing the GWAS may also contribute to discrepancies between the estimated LD structure.

## Computational efficiency

BEAVR uses Gibbs sampling [20] to estimate the posterior probability of the regional polygenicity parameter. A naive implementation of the Gibbs sampler has a per-iteration computational complexity of $\mathcal{O}(M_r^2)$, where $M_r$ is the number of SNPs in the region. By leveraging the expected sparsity of the causal status at each SNP, we can improve the run-time of the algorithm to $\mathcal{O}(M_r K_r)$, where $K_r$ is the number of causal SNPs in the region. Fig 4A shows that this improvement leads to a 12-fold improvement in run-time for a region with 5, 000 SNPs. To assess how the number of causal SNPs affects the efficiency of our algorithm, we generated

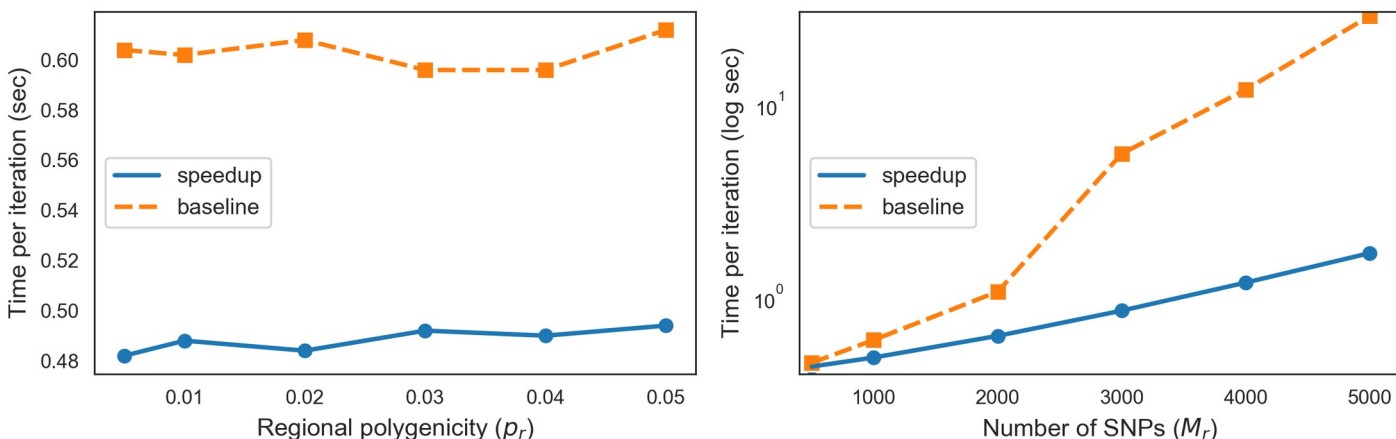

**Fig 4. BEAVR is computationally efficient. (A)** We show the run-time in terms of seconds per iteration of the Gibbs sampler (log-scale). We compare the version of BEAVR with the algorithmic speedup outlined in Materials and methods ('speedup') versus the straightforward implementation ('baseline'). We vary the number of SNPs in the region while fixing the polygenicity of each region to $p_r = 0.01$. **(B)** We show the runtime of the sampler when the number of SNPs in the region is fixed to $M = 1, 000$ and we vary the polygenicity.

**Table 1. Genome-wide estimates of polygenicity and total SNP heritability.**

| Trait | %regions | $h^2_{GW}$ | p | $M_c$ | p-*GENESIS* |
|---|---|---|---|---|---|
| *BMI* | 66.2 | 0.30 (0.004) | (0.017, 0.018) | $(7.67 \times 10^3, 8.41 \times 10^3)$ | 0.014(0.003) |
| *Height* | 79.6 | 0.64 (0.004) | (0.030, 0.032) | $(1.37 \times 10^4, 1.45 \times 10^4)$ | 0.009(0.002) |
| *Waist-hip ratio* | 40.6 | 0.18 (0.004) | (0.007, 0.008) | $(3.12 \times 10^3, 3.57 \times 10^3)$ | 0.009(0.003) |
| *Diastolic blood pressure* | 35.5 | 0.16 (0.004) | (0.006, 0.007) | $(2.54 \times 10^3, 3.01 \times 10^3)$ | – |
| *Systolic blood pressure* | 34.9 | 0.17 (0.004) | (0.006, 0.007) | $(2.58 \times 10^3, 3.01 \times 10^3)$ | – |

We report the percentage of 6-Mb regions containing at least one causal SNP under the column '% regions'. Genome-wide estimates of polygenicity and heritability were computed by aggregating estimates across all regions. The standard error is reported for genome-wide heritability estimates. Here, *p* denotes the proportion of causal SNPs and $M_c$ denotes the total number of causal SNPs (we report the 95% posterior credible interval for each of these parameters). The last column denotes the estimates of polygenicity computed in a previous study [2] (twice the standard error is reported in parentheses). We omit estimates for traits not available in the prior study.

simulated GWAS data for 1,000 SNPs and varied the regional polygenicity from $p_r = 0.005$, 0.01, 0.02, 0.03, 0.04, 0.05 and observe efficiency gains across the range of parameters (Fig 4B). The optimization of our method makes it possible to efficiently analyze regions of various sizes as well as densely imputed regions with thousands of variants.

## Contrasting genome-wide and regional polygenicity across complex traits

We applied BEAVR to estimate regional polygenicity from marginal effect size estimates for five anthropometric and blood pressure traits from the UK Biobank (see Table 1). Marginal association statistics were computed for each of these traits from a subset of unrelated individuals identified as White British (see Materials and methods). We applied BEAVR by dividing the genome into a total of 470 6-Mb regions where each region has on average 1, 000 SNPs. Since BEAVR requires an estimate of LD between the SNPs, we used in-sample LD, *i.e.*, LD computed on the White British subset of the UK Biobank. We additionally used HESS [12] to estimate regional heritability. Since BEAVR produces a posterior distribution of the regional polygenicity, we report a region to have nonzero polygenicity if the posterior mean—(2× posterior standard deviations) does not overlap 0. Furthermore, we estimate the genome-wide polygenicity for a trait as the sum of the posterior means of regional polygenicity across all regions.

Consistent with previous estimates of genome-wide polygenicity [2], we observe that all the analyzed traits are highly polygenic. Across the traits, we observe that over one-third of the regions in the genome contain at least one causal SNP and overall each of the traits is estimated to harbor at least 1, 000 causal SNPs (Table 1). We also observe variation across traits: for height, nearly 80% of the regions contain at least one causal SNP and the total number of causal SNPs could be as high as 15, 000 while blood pressure traits are estimated to harbor about 2, 500 − 3, 000 causal SNPs. Our estimates for the proportion of causal SNPs for height is significantly higher than previously reported [2] (Table 1): the 95% credible interval estimated by BEAVR is (3.0%, 3.2%) while the estimates from prior work [2] are 0.9% with standard error 0.1%. We hypothesize that this difference is due, in part, to our method capturing smaller effect sizes by fully modeling LD, which is consistent with our simulations, but could also arise from the differences in SNP sets and GWAS summary statistics analyzed.

Previous studies have used the proportion of genomic regions with nonzero heritability as a proxy for polygenicity since nonzero heritability requires at least one causal SNP in the region [12, 21]. However, the distribution of regional heritability does not fully reflect the distribution of regional polygenicity (Fig 5, S4 and S5 Figs). Across the traits, the proportion of regions

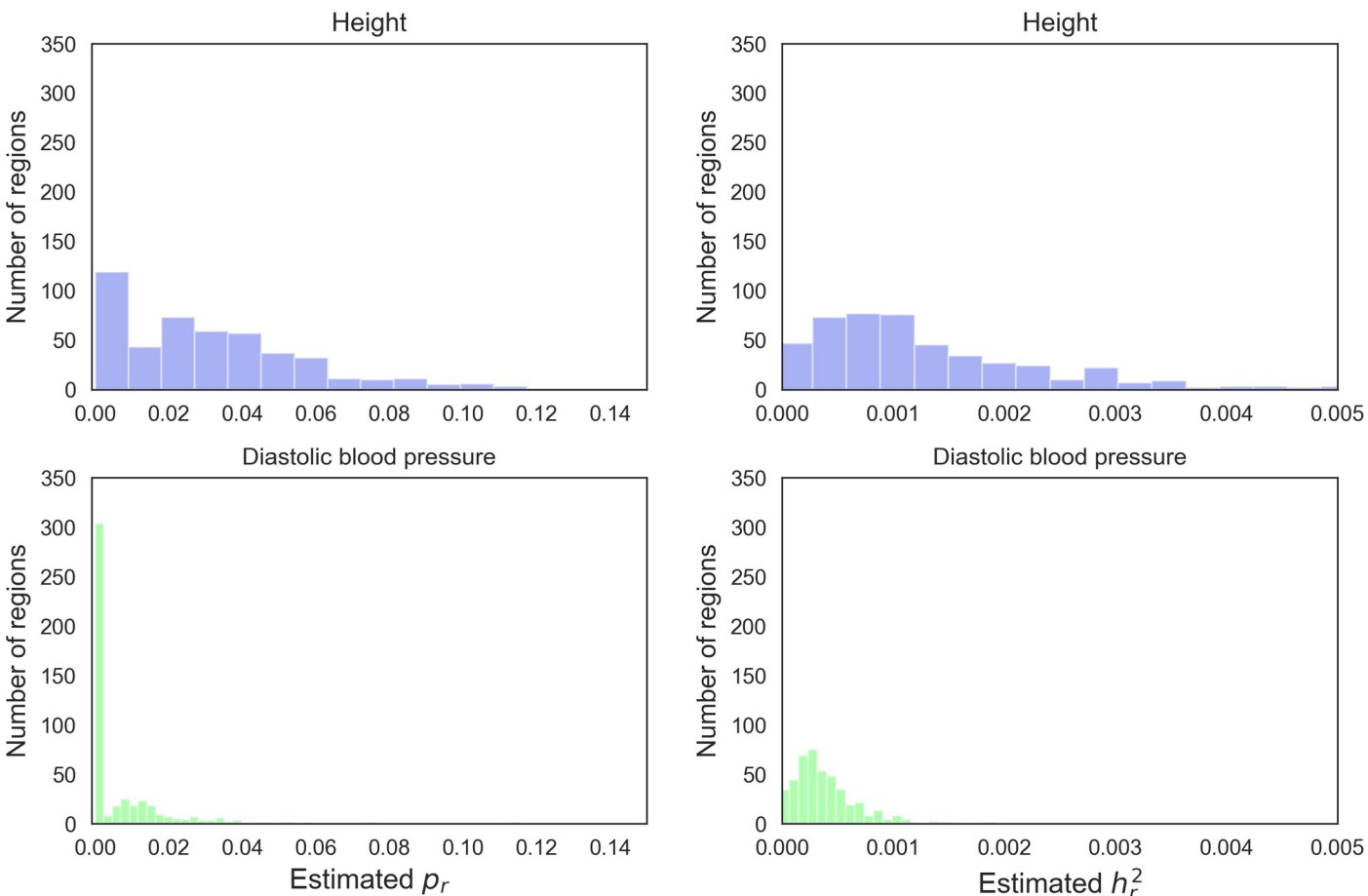

**Fig 5. Distribution of regional polygenicity and heritability.** We divide the genome into 6-Mb regions and report the posterior mean of the regional polygenicity for each region across height and diastolic blood pressure. Using external software [12], we report the distribution of regional heritability for each trait.

containing at least one causal SNP is substantially higher than the estimated proportion of causal SNPs across the genome (Table 1): while ≈ 80% of regions contain at least one causal SNP for height, we estimate that ≈ 3% of the SNPs are casual. Further, we observe wide variation in regional polygenicity where, across the analyzed traits, nearly 50% of regions contain at least 5 causal SNPs and about 5% of regions contain at least 50 causal SNPs (Fig 5 and S5 Fig). These results demonstrate the additional information that can be obtained from estimates of regional polygenicity.

## Heritability is proportional to the number of causal SNPs

Previous studies have documented a linear relationship between chromosome length and the per-chromosome heritability for multiple traits suggesting that the architecture of these traits is highly polygenic [12, 31]. We replicate this relationship between the number of SNPs and the heritability in a genomic region for each trait ($p$-value = $1.22 \times 10^{-13}$; $R^2$ = 0.162 averaged across traits; Table 2). In addition, we observe that a linear regression of heritability on the number of causal SNPs in the region is significant ($p$-value = $2.60 \times 10^{-21}$; $R^2$ = 0.278 averaged across traits) (Table 2). We also observe that the number of causal SNPs in a region better explains regional heritability than the number of overall SNPs in the region. This ranges from approximately the same $R^2$ in systolic blood pressure to nearly three times in WHR (Table 2).

**Table 2. Linear relationship between heritability, number of SNPs, number of causal SNPs, and genomic annotations.**

| Trait | $R^2(h_r^2 \sim M_r)$ | $R^2(h_r^2 \sim M_{C_r})$ | $R^2(h_r^2 \sim M_r + M_{C_r})$ | $R^2(h_r^2 \sim \text{all-annotations})$ |
|---|---|---|---|---|
| BMI | $0.182 \ (3.67 \times 10^{-22})$ | $0.226 \ (7.64 \times 10^{-28})$ | $0.347 \ (6.38 \times 10^{-44})$ | $0.532 \ (1.20 \times 10^{-27})$ |
| Height | $0.172 \ (5.34 \times 10^{-21})$ | $0.447 \ (2.96 \times 10^{-62})$ | $0.501 \ (3.30 \times 10^{-71})$ | $0.647 \ (3.23 \times 10^{-47})$ |
| Waist-hip ratio | $0.105 \ (6.13 \times 10^{-13})$ | $0.295 \ (2.05 \times 10^{-37})$ | $0.352 \ (1.08 \times 10^{-44})$ | $0.540 \ (8.23 \times 10^{-29})$ |
| Diastolic blood pressure | $0.183 \ (2.60 \times 10^{-22})$ | $0.254 \ (1.08 \times 10^{-31})$ | $0.359 \ (7.33 \times 10^{-46})$ | $0.530 \ (2.09 \times 10^{-27})$ |
| Systolic blood pressure | $0.168 \ (2.09 \times 10^{-20})$ | $0.169 \ (1.30 \times 10^{-20})$ | $0.311 \ (1.77 \times 10^{-38})$ | $0.532 \ (1.11 \times 10^{-27})$ |

In the first column, we model the linear relationship between the heritability of a trait and the number of SNPs across all regions of the genome. We report the coefficient of determination ($R^2$). The relationship is significant for all traits (*p*-values are reported in parentheses). We observe a similar trend relating the heritability and number of causal SNPs in a region. We perform a multivariate regression to assess the relationship between the heritability and both the number of SNPs and number of causal SNPs in a region. Finally, in the last column, we perform a multivariate regression of heritability on the number of SNPs, number of causal SNPs, number of genes, median *B* value, and functional annotations [28]. Significant annotations are listed in Table B in S1 File.

The slope of the regression of regional heritability on the number of causal SNPs averaged across traits is $1.63 \times 10^{-5}$, which can be interpreted as the heritability per additional causal SNP (Table A in S1 File). Performing multiple regression, we find that both the number of SNPs and number of causal SNPs have a significant relationship to the heritability in a region (average *p*-value = $3.60 \times 10^{-39}$; $R^2 = 0.374$). We hypothesize that the number of SNPs and causal SNPs together explain more of the variation in heritability than the number of causal SNPs alone due, in part, to inaccurate estimates of the number of causal SNPs and regional heritability as well as possible misspecifications in the model assumed by BEAVR.

We further investigate the relationship between genomic annotations and heritability as well as the number of causal SNPs in a region. Including the number of genes, median *B* value, and functional annotations [28] as covariates in the regression (see Materials and methods), only the number of causal SNPs remains significant (average *p*-value = $6.37 \times 10^{-11}$, *p*-value < 0.05/(number of annotations)) while the total number of SNPs in the region remains significant for 3 out of 5 traits (Table B in S1 File). None of the other genomic annotations are significant after the multiple testing correction.

While the expected regional heritability can be partly explained by the number of causal SNPs, we also observe regions that have disproportionately high heritability given the number of estimated causal SNPs (Fig 6 and S6 Fig and Table A in S1 File). These outlier regions (defined as regions with an absolute studentized residual larger than 3) are likely to harbor SNPs with larger effect sizes compared to other regions. Consistent with this hypothesis, 24 out of 27 outlier regions contain at least one genome-wide significant SNP for at least one trait. This proportion is significantly higher than a randomly chosen set of 27 regions (*p*-value < $\frac{1}{1,000}$). Taken together, our analyses indicate that the heritability of a trait is composed of a mixture of small effect SNPs as well as some SNPs with relatively larger effects.

Finally, we also investigate whether the gene density in a region plays a role in the observed regional polygenicity estimates. We perform a likelihood ratio test between the following two models to assess the effect of gene density on the number of causal SNPs ($M_{C_r}$) after adjusting for both regional heritability and the number of SNPs:

$H_0 : M_{C_r} \sim h_r^2 + M_r; H_1 : M_{C_r} \sim h_r^2 + M_r + \#\text{genes}$. As shown in Table C in S1 File, we find that only the likelihood ratio test for height is significant after adjusting for the number of tested traits (*p*-value < $\frac{0.05}{5}$). This observation could be due to the fact that we included all protein coding genes in the analysis regardless of the specific biological mechanism of each gene. For example, when analyzing BMI, one would expect regions with genes related to

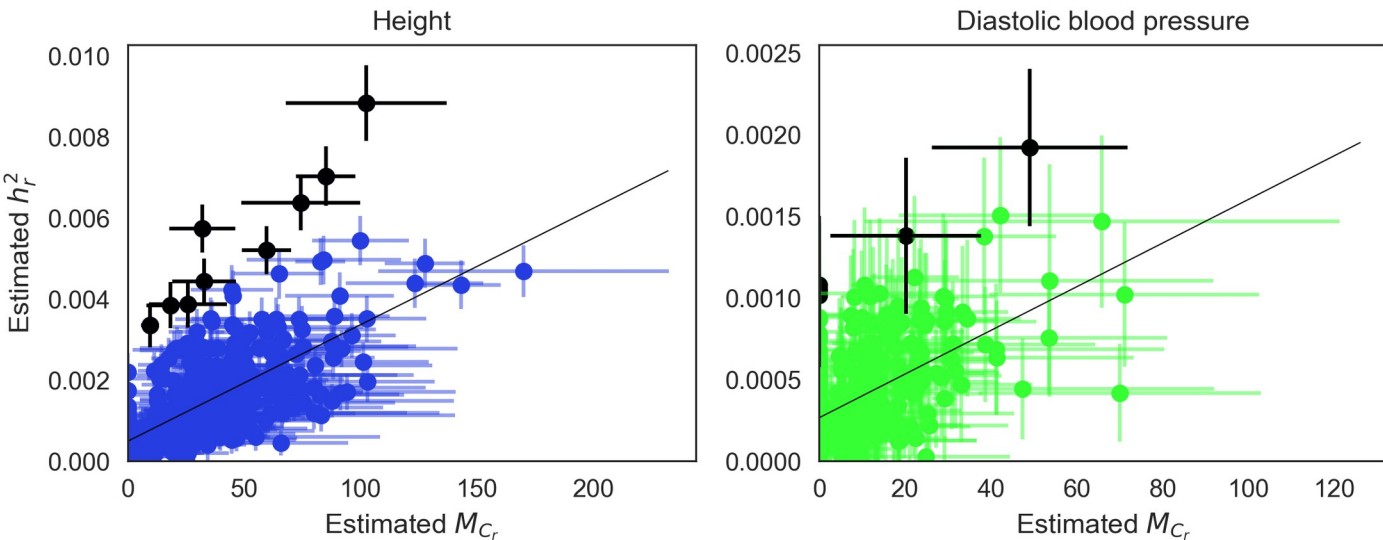

**Fig 6. Heritability is proportional to the number of causal SNPs in a region.** We show the relationship between the number of causal SNPs and heritability for each region across height and diastolic blood pressure. We fit a linear regression for each trait and report the slope of the regression, which can be interpreted as the increase of heritability per additional causal SNP. Horizontal error bars represent two posterior standard deviations around our estimates for the number of causal SNPs. Vertical error bars represent twice the standard error around the estimates of regional heritability. Dots in black denote outlier regions which have an absolute studentized residual larger than 3.

lipids or metabolism to harbor more causal variants than genes related to seemingly unrelated biological mechanisms. This observed effect of gene density on the polygenicity of height is consistent with the hypothesis that genes throughout the genome, regardless of the specific biological function, contribute to the variance of height. Previous work has shown that height is one of the most polygenic traits with numerous causal variants spread throughout the genome [15]. This idea that numerous genes, regardless of functional mechanism, have an effect on a trait is related to the recently proposed 'omnigenic' model [5].

## Discussion

In this work, we propose BEAVR, a novel, scalable method to estimate regional polygenicity from GWAS effect size estimates in a Bayesian framework. We employ a fast inference algorithm that enables efficient inference while fully accounting for LD. Applying BEAVR to anthropometric and blood pressure traits in the UK Biobank, we observe that all of the analyzed traits are highly polygenic. At least a third of 6-Mb regions harbor at least one causal variant with this fraction rising as high as 80% for height. We find that the proportion of regions containing at least one causal SNP, which is often used as a proxy for polygenicity in previous studies, is much higher than our estimates of the proportion of causal SNPs. Additionally, we observe wide variation in regional polygenicity with an average of 48.9% of regions across the analyzed traits containing at least 5 causal SNPs and 5.44% of regions containing at least 50 causal SNPs. Finally, we find that the number of causal SNPs better explains variation in SNP heritability across regions compared to the total number of SNPs.

The observed polygenic architecture of complex traits supports the hypothesis that the majority of trait variation is modulated by variants distributed across the genome. Trait heritability is largely driven by the number of causal variants and most of these variants are spread uniformly across the genome. This finding suggests that a large proportion of genes have at least some, although limited, effect on a trait. These findings are consistent with the recently

proposed omnigenic model which suggests that disease risk is driven by a combination of a small number of primary 'core' genes and numerous 'peripheral' genes which are connected to core genes via highly interconnected gene networks [5].

We conclude by discussing limitations of our study and directions for future work. First, our model assumes that the causal effects are drawn from a single Gaussian distribution. This assumption can be relaxed and other distributions (such as mixtures of Gaussians) can be used instead. Second, our estimates of genome-wide polygenicity assume that the LD matrix is block structured which allows us to estimate genome-wide polygenicity by applying our method to regions corresponding to LD blocks. Finally, our analyses in the UK Biobank were limited to array data and thus the set of SNPs used in our analyses are missing true causal SNPs that were not typed. We leave a more thorough investigation of this scenario and analyses on imputed data as future work.

## Supporting information

**S1 Fig. A graphical model describing the generative process of our data.** Directed graphical model diagram for BEAVR.
(TIF)

**S2 Fig. Inferring regional polygenicity from non-Gaussian causal effect size distributions.** We simulate effect sizes from mixture of Gaussian distributions with the following set of variance components: $[1 \times 10^{-3}, 1 \times 10^{-4}]$; $[1 \times 10^{-4}, 1 \times 10^{-5}]$; $[1 \times 10^{-3}, 1 \times 10^{-4}, 1 \times 10^{-5}]$. The polygenicity of the region equals the sum of the mixture proportions and the number of causal SNPs are spread equally amongst all the mixture components. For causal effect sizes drawn from the distribution with larger variances (e.g. $1 \times 10^{-3}$), our estimates are relatively unbiased. However, for distributions with smaller variance components (e.g. $1 \times 10^{-5}$), we start to see a downward bias proportional to the fraction of SNPs drawn from the distribution with the smaller variance component(s).
(TIF)

**S3 Fig. Assessing the role of reference LD in estimating regional polygenicity.** The first GWAS (left) is simulated LD computed with genotypes from the UK Biobank ($N = 337, 205$) and inference is performed using LD computed from the European individuals from the 1000 Genomes reference panel ($N = 503$). The second GWAS (right) is simulated with LD derived from a subset of ($N_1 = 168, 602$) genotypes from the UK Biobank and inference is performed using LD computed from a separate, non-overlapping subset ($N_2 = 168, 602$) of individuals also from the UK Biobank. We find that when using LD from separate studies (1000 Genomes), BEAVR fails to accurately estimate the regional polygenicity. However, when we use LD computed from a separate set of individuals from the same study, we find our estimates are approximately unbiased.
(TIF)

**S4 Fig. Distribution of regional heritability.** We divide the genome into 6-Mb regions and report the posterior mean of the regional polygenicity for each region across BMI, waist-hip ratio (WHR), and systolic blood pressure.
(TIF)

**S5 Fig. Distribution of regional polygenicity.** Using external software [12], we estimate the heritability in each 6-Mb region for each trait. We report the distribution of regional heritability for BMI, waist-hip ratio (WHR), and systolic blood pressure.
(TIF)

**S6 Fig. Heritability is proportional to the number of causal SNPs in a region.** We show the relationship between the number of causal SNPs and heritability for each region across BMI, waist-hip ratio (WHR), and systolic blood pressure. We fit a linear regression for each trait. Horizontal error bars represent two posterior standard deviations around our estimates for the number of causal SNPs. Vertical error bars represent twice the standard error around the estimates of regional heritability. Dots in black denote outlier regions which have an absolute studentized residual larger than 3.
(TIF)

**S1 File. Supplementary materials.** Additional derivations for the Gibbs sampler. **Table A. Linear relationship between the number of causal SNPs and heritability**. We model the linear relationship between the number of causal SNPs for a trait and the heritability across all regions of the genome. We report the slope of the regression and the standard error. The slope can be interpreted as the expected per-SNP heritability contribution per causal SNP. The last column reports the number of 'outlier' regions for each trait, defined as a region with an absolute studentized residual greater than 3. **Table B. Covariates that are associated with regional heritability $h_r^2$**. We perform a multivariate regression of heritability on the number of SNPs, number of causal SNPs, number of genes, median $B$-statistic, and non-cell-type-specific annotations [28]. Only the number of causal SNPs ($M_{C_r}$) remains significant for all traits after the multiple testing correction (average $p$-value = $6.37 \times 10^{-11}$), and the number of SNPs ($M_r$) remains significant for 3 our of 5 traits after the multiple testing correction. **Table C. Likelihood ratio test assessing the role of gene density in regional polygenicity estimates**. We perform a likelihood ratio test between the following two models to assess the effect of gene density on the number of causal SNPs ($M_{C_r}$) after adjusting for both regional heritability and the number of SNPs ($H_0 : M_{C_r} \sim h_r^2 + M_r; H_1 : M_{C_r} \sim h_r^2 + M_r + \#\text{genes}$).
(PDF)

## Acknowledgments

We are grateful to Nicholas Mancuso, Rob Brown, Alec Chiu for their helpful and insightful discussions.

## Author Contributions

**Conceptualization:** Bogdan Pasaniuc, Sriram Sankararaman.

**Data curation:** Kathryn S. Burch, Kangcheng Hou.

**Formal analysis:** Ruth Johnson, Kangcheng Hou, Mario Paciuc.

**Funding acquisition:** Bogdan Pasaniuc, Sriram Sankararaman.

**Investigation:** Ruth Johnson, Mario Paciuc.

**Methodology:** Ruth Johnson, Sriram Sankararaman.

**Software:** Ruth Johnson.

**Supervision:** Bogdan Pasaniuc, Sriram Sankararaman.

**Validation:** Ruth Johnson, Mario Paciuc.

**Writing – original draft:** Ruth Johnson, Kathryn S. Burch, Bogdan Pasaniuc, Sriram Sankararaman.

**Writing – review & editing:** Ruth Johnson, Kathryn S. Burch, Bogdan Pasaniuc, Sriram Sankararaman.

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
