## [Decision Letter · Decision Letter 0]

21 Jun 2021

Dear Dr. Sankararaman,

Thank you very much for submitting your manuscript "Estimation of regional polygenicity from GWAS provides insights into the genetic architecture of complex traits" for consideration at PLOS Computational Biology.

As with all papers reviewed by the journal, your manuscript was reviewed by members of the editorial board and by several independent reviewers. In light of the reviews (below this email), we would like to invite the resubmission of a significantly-revised version that takes into account the reviewers' comments.

We cannot make any decision about publication until we have seen the revised manuscript and your response to the reviewers' comments. Your revised manuscript is also likely to be sent to reviewers for further evaluation.

Sincerely,

Heather E. Wheeler, Ph.D.

Guest Editor

PLOS Computational Biology

Sushmita Roy, Ph.D.

Deputy Editor

PLOS Computational Biology

Reviewer's Responses to Questions

**Comments to the Authors:**

Reviewer #1: Major:

1) Figure 2 results I find troublesome. I presume that when p_r has downward bias, so will the estimation of heritability. and if the posterior inclusion probabilities were used as evidence for association, BEAVR will have low power for typical GWAS datasets whose sample sizes are far less than 50k. Although one may argue that the main goal of BEAVR is to estimate polygenicity, but my opinion is that effect sizes, heritability, and polygenicity are intimated connected, and it would be different for me to believe it does one better while others worse.   Also, why BEAVR does well extremely large samples? My guess is that the poster BEAVR sampled is narrowly picked, so that even a naively implemented sampling scheme can get it right with relatively small amount of samples.

2) The authors claim their approach “yields approximately unbiased estimates of regional polygenicity.” This begs the question how does the ascertainment bias on genetic variants affect the estimation of polygenicity. For example, if SNPs of minor allele frequency of 1% were removed, will the polygenicity estimate be similar to SNP maf cutoff of 5%?

3) How to obtain regional heritability? Estimated by HESS. You may reestimate the regional heritability after you fit the BEAVR, how does it compare to the one estimated by HESS?

Minor:

1) The word “causal” was used casually by the authors.

2) “Heritability enrichments are largely driven by variation in the number of causal SNPs.” Is this consistent with omnigenic model?

3) Clarify the definition of region independent, and how to account for the different number of variants in each region.

4) Bayesian estimation of variants in a region, shouldn’t the acronym be BEVAR instead of BEAVR?

5) Figure 3. why in (b) larger M has smaller variation? is that because large M implies less correlation between a random pair of SNPs?

6) Line 178. “We additionally used HESS to estimate regional polygenicity.” Is this a typo? Should polygenicity be heritability?

7) Does regional polygenicity has anything to do with gene density in the region? You may compare regions on chr13 with regions on chr19.

8) Line 376. “We ran BEAVR for 1000 iterations with a burn-in of 250 iterations”. These numbers appears to be too small. How many proposals / updates in each iteration? How is consistency between independent runs?  

9) The novel elements in computation need to be presented more clearly and convincingly.

Reviewer #2: The work by Johnson et al aims to accurately and systematically estimate regional polygenicity for different traits through bayesian inference on the effect sizes using a spike and slab prior. Through simulations authors show that their approach has low bias in a standard GWAS setting and yields precise estimates and is computationally scalable. The authors show that their systematic approach reveals that there is substantial heterogeneity in regional polygenicity for a given trait in UKBB and produces a more interpretable estimate as compared to previous proxy estimates that were used. I have a few minor points which might make the work stronger:

1. Although the authors mention that in-sample LD is used throughout, I believe it is critical to understand what happens when reference panel LD estimates are used. Although GWAS summary statistics are widely available, the corresponding in-sample LD sharing is not a practice that is very common. So, for BEAVR to widely applicable, I believe the authors should investigate or at least address in a basic user-case-scenario the problems one might expect when in sample LD is not available and LD needs to be estimated from reference panel, and how misspecification affects the inference.

2. Can the authors provide any intuition as to why the inference does not depend at all on \\alpha parameter?

3. For highly unbalanced case-control ratio disease phenotypes, the effect sizes/scores might not follow a Gaussian distribution. Since normality is an assumption, can the authors state as to how deviations from normality distorts the estimates? Is the method still valid ?

**Have the authors made all data and (if applicable) computational code underlying the findings in their manuscript fully available?**

Reviewer #1: Yes

Reviewer #2: None

PLOS authors have the option to publish the peer review history of their article (what does this mean?). If published, this will include your full peer review and any attached files.

Reviewer #1: No

Reviewer #2: No
---

## [Decision Letter · Decision Letter 1]

27 Sep 2021

Dear Dr. Sankararaman,

We are pleased to inform you that your manuscript 'Estimation of regional polygenicity from GWAS provides insights into the genetic architecture of complex traits' has been provisionally accepted for publication in PLOS Computational Biology.

Best regards,

Heather E. Wheeler, Ph.D.

Guest Editor

PLOS Computational Biology

Sushmita Roy

Deputy Editor

PLOS Computational Biology

Reviewer's Responses to Questions

**Comments to the Authors:**

Reviewer #1: The authors have done a good job to address my prior concerns.

I have only one minor point need the author to clarify.

To address minor point 8, authors stated: "We utilize a simple heuristic to initialize all of our parameters in the Gibbs sampler..." A Gibbs Sampler acceptance rate is alway 1. What does the authors mean by "The average acceptance proposal (computed across 100 runs) is 3.8% (sd: 0.86%)." ?

Reviewer #2: The authors have done a commendable job to addressed my comments satisfactorily. I thank them for that. I have no further comments.

**Have the authors made all data and (if applicable) computational code underlying the findings in their manuscript fully available?**

Reviewer #1: Yes

Reviewer #2: None

PLOS authors have the option to publish the peer review history of their article (what does this mean?). If published, this will include your full peer review and any attached files.

Reviewer #1: No

Reviewer #2: No

---

## [Editor Report · Acceptance letter]

18 Oct 2021

PCOMPBIOL-D-21-00755R1

Estimation of regional polygenicity from GWAS provides insights into the genetic architecture of complex traits

Dear Dr Sankararaman,

I am pleased to inform you that your manuscript has been formally accepted for publication in PLOS Computational Biology. Your manuscript is now with our production department and you will be notified of the publication date in due course.

With kind regards,

Agnes Pap
